# The Influence of the Clinicians’ Experience on the Outcome of Dental Implants: A Clinical Audit

**DOI:** 10.3390/healthcare11152201

**Published:** 2023-08-04

**Authors:** Omir Aldowah, Hamad Alawad, Mohammad Alqhtani

**Affiliations:** 1Prosthetic Dental Science Department, Faculty of Dentistry, Najran University, Najran 66462, Saudi Arabia; aldowah@gmail.com; 2Lab of Aseer Central Hospital, Ministry of Health, Riyadh 11595, Saudi Arabia; halawad@moh.gov.sa

**Keywords:** clinician experience, success rate, survival rate, risk factors

## Abstract

The purpose of this outcome audit is to evaluate the influence of the clinicians’ experience on the outcome of dental implants. In addition, it is to identify the associated risk factors that might influence the success and survival of these implants. Methodology: The records of patients treated with SLA/SLActive Straumann implants were screened. This enabled us to have a minimum of 12 months of follow-up. Eligible patients, according to the inclusion criteria, were contacted and invited to undergo a follow-up assessment. Success was accounted for and defined in a comprehensive manner by considering four different categories: implant perspective, peri-implant soft tissue perspective, prosthetic perspective, and patient satisfaction. The patient investigations included a clinical examination of the implant mobility, suppuration, width of keratinized mucosa, probing depth, plaque accumulation, prosthetic complications, and patient satisfaction. In addition, a periapical radiograph was taken to evaluate bone loss and peri-implant radiolucency. The data were analysed using SPSS version 26. Results: Thirty-eight patients with 84 SLA/SLActive Straumann implants were available for the assessment. The mean age of the patients at implant surgery was 49.05 ± 13.19 years. Over the mean follow-up period of 26 months, no implant fractures were noted. Overall, eight implants were considered failures (9.5%). Two out of six patients with a history of periodontitis (HoP) and two out of five smokers exhibited failed implants. The patients’ satisfaction responses showed that all the responses were statistically higher than the test median value of three. The median value of general satisfaction using a visual analogue scale was 9 out of 10. Conclusions: The implants placed on partially and fully edentulous patients revealed high survival and success rates (100% and 90.5%, respectively) at a mean follow-up time of 26 months. It can be concluded that the implant practise among trainees in the programme is satisfactory. A history of periodontitis and a lack of patient compliance with supportive periodontal therapy in some cases have been shown to be risk factors associated with increased implant failure, mainly peri-implantitis.

## 1. Introduction

Several studies have demonstrated high implant success and survival rates of more than 90%, and sometimes up to 100%, depending on their follow-up time and defined success criteria [1,2,3,4,5]. Generally, long-term outcomes of dental implants surgically and restoratively implanted could be weighted to the outcomes of other treatment options [6,7,8]. Moreover, several studies have assessed implant success based on the bone level [9,10,11,12], and other studies have considered more aspects of the implant–prosthetic complex, in addition to patient satisfaction as a whole to assess success criteria [13,14,15].

A learning curve is defined as the rate or course of the progress required to learn something new [16]. The influence of the learning curve of the operator on the outcome of dental implants is a controversial issue. Lambert et al. claimed that implants placed by surgeons who had placed fewer than 50 implants (less experienced) failed twice as frequently as implants inserted by surgeons who had placed more than 50 implants (more experienced) [17]. However, the authors did not explain how they had arrived at the number 50 as the scale for surgeon experience, which made the validity of this number uncertain. In addition, caution should be shown when interpreting this retrospective study due to the unclear methodology, multiple confounders, and unknown protocol of the implant placement and loading. A similar conclusion was drawn by Preiskel and Tsolka [18]. In contrast, other studies have asserted that the survival rates for dental implants inserted by trainee surgeons are comparable to the survival rates reported in the literature. They have demonstrated that surgeon experience does not affect implant survival and found a survival rate of 96% for implants performed by students [19,20]. The few available studies on dental implant outcomes from postgraduate trainees do not include a comprehensive assessment of success criteria. However, by assessing the implant treatment provided by postgraduate trainees, the outcomes could be compared to the available evidence and provide better understanding of the operator factors that influence these outcomes [21].

The purpose of this outcome audit is to evaluate the influence of the clinicians’ experience on the outcome of dental implants. In addition, it is to identify the associated risk factors that might influence the success and survival of these placed implants.

## 2. Materials and Methods

Following most recent systematic review of the long-term success and survival of dental implants [22], the standard for this clinical audit was set at 90% for success rate and 95% for survival rate.

The recruited patients were treated by postgraduates doing a Masters of Clinical Dentistry (MClin Dent) in the prosthodontics programme at the Cardiff University Dental Hospital, United Kingdom. The trainees had not placed implants prior to their enrolment in this postgraduate programme. All the included 17 trainees had equal implant experience and the same chances of training. The case selection for implant treatment at the (MClin Dent) excludes patients with active periodontitis, uncontrolled diabetes mellitus, heavy smoking (more than 20 cigarettes per day) [23], and those using intravenous bisphosphonate for treating osteoporosis. All the implants had SLA or SLActive surface (Straumann Intl., Basel, Switzerland). 

Dental implants for which there were postoperative periapical radiographs available at placement time and those that had a follow up for a minimum of 12 months after placement of restoration were included in this study. The participants signed a written consent form after being informed in detail about the objectives and methods of the study. Patient records were revised. Data relating to the patients, implants, and prostheses were retrieved from patients’ medical files and updated during clinical examination. 

The following measurements were recorded at a minimum of 12 months after the loading of prosthesis. The recorded data included parameters in relation to implant prospective and peri-implant tissue, such as implant mobility, modified plaque indexed (mPI), modified bleeding indexed (mBI), width of keratinized mucosa, suppuration, and peri-implant radiolucency. 

The clinical parameter of pain was evaluated subjectively by asking the patients whether they experienced pain in addition to percussion evaluation. Mobility and pus discharge were examined clinically using two instruments and the naked eye for pus discharge. 

A radiographic assessment was performed using a paralleling technique to evaluate marginal bone loss (MBL) and peri-implant radiolucency. 

The width of the keratinised mucosa was measured using a William probe. A peri-implant probing measurement and bleeding upon probing were performed with a plastic periodontal probe. The scoring was based on a mPI and mBI, as described by Mombelli et al., from the crowns or implant abutment at four points [24]. All the measurements were performed at 4 points per implant: mesiobuccal (MB), midbuccal (B), distobuccal (DB), and midlingual (L). 

All the clinical and radiographic assessments were performed by one previously calibrated examiner (O.A) who is a postgraduate student. The proper angulation of the periapical radiographs was determined by a correct illustration of the implant threads, Figure 1. 

The prostheses were classified by their design as follows: single crown, fixed partial denture, and implant-supported overdenture.

The implant patient satisfaction questionnaire described by Pjetursson et al. was modified to fit this study and include implant maintenance, Table 1 [25].

The participants determined their satisfaction through a visual analogue scale (VAS) consisting of a 100 mm straight line, with the left end denoting extreme satisfaction and the right end extreme dissatisfaction. This scale measurement was converted into 5 nominal categories as follows: very satisfied (81–100 mm), satisfied (61–80 mm), neutral (41–60 mm), dissatisfied (21–40 mm), and not at all satisfied (0–20 mm).

In this audit, it was attempted to evaluate the comprehensive success criteria for implants to consider the result of the audit with some degree of clinical validation. However, it seems that the success rate will consistently decrease because of the increase in the number of parameters included for the assessment of success. The four most commonly used criteria for evaluating outcomes with dental implants in different studies are implant fixtures, peri-implant tissues, prostheses, and subjective patient evaluation, as shown in Table 2 [26]. The recorded objective data included parameters in relation to implant level and peri-implant tissue, such as implant mobility, mPI, mBI, width of keratinized mucosa, suppuration, and peri-implant radiolucency. At the prosthetic level, the patients’ notes were checked for any complications at any time from the implant placement until the day of the follow-up visit. Complications were categorised as biological/technical and new/recurrent. The severity for each complication was graded as mild, moderate, or severe, and the complication was graded as managed or on-going. Subjective parameters involving the patients’ satisfaction with aesthetic, function, ability to taste, and general satisfaction were graded as very satisfied, satisfied, neutral, dissatisfied, or not at all satisfied. Success was accounted for and defined by the presence of the following parameters, according to Gallucci et al. [27]:No implant mobility and no signs of constant infection, discomfort, or radiolucency [28].Bone loss at first year less than 1.5 mm and annual bone loss less than 0.2 mm thereafter.A mean mPI and mBI value of 1 or less.A mean probing depth of less than 5 mm.Preservation of at least 1.5 mm of buccal/lingual keratinised mucosa.Four or less complications of a mild or moderate severity in prostheses that were terminated.

The overall patient satisfaction with treatment was rated as either good or excellent [29].

Survival was defined as implants or prostheses that did not need to be replaced.

Complications at the prosthetic level were searched in the patients’ records and updated clinically. The complications were categorised to new or recurrent. The severity for each complication was graded as mild, moderate, or severe, and the complication was graded as managed or on-going. 

The statistical software program SPSS v.26 (Chicago, IL, USA) was used for the statistical analyses, and descriptive statistics were reported for the continuous variables. In the questionnaire on satisfaction, median values were tested against the test median value of 3 using a non-parametric sign and Wilcoxon signed-rank test. A simple t-test was used for patient satisfaction, and the alpha was set at 0.05. An intra-examiner reliability test was performed. To conduct this intra-examiner reliability test, 10% of the samples were selected randomly. Then, measurements were recorded for the second time and analysed using SPSS v.26 (Chicago, IL, USA). 

## 3. Results

### 3.1. Intra-Examiner Reliability Test

The analysis revealed that the single measure of the intra-class correlation was 0.862 and the average measure was 0.926; this is considered good. 

### 3.2. General Descriptive Result

In total, 38 patients (84 implants) were included in this study out of 73 patients (151 implants). A total of 23 patients were female (60.5%) and 15 were male (39.5%). The mean age of the patients was 49.05 ± 13.19 years. Half of the implants were placed in the anterior maxilla. 

Eighty-four (84) implants were inserted in those thirty-eight patients. Table 3 shows the distribution of the implant sites. 

Table 3 and Table 4 represent the distribution of the size and length of the used implants. The distribution of the prostheses is shown in Table 5.

### 3.3. Success and Survival Rates

The success and survival rates of the implants considered were 90.5% (n = 76) and 100% (n = 84), respectively, at a mean follow-up time of 26 months.

### 3.4. Implant Level

In the follow-up, the parameters at the implant level showed that no patient complained of pain or their implants being mobile, lacking a solid sound upon tapping, or showing peri-implant radiolucency. Based on radiograph radiography, the mean of the bone loss in the 84 implants was 1.22 ± SD 0.90 mm (range: 0.0–4.3). Eight implants (9.5%) were considered as failed, as they exceeded the accepted bone loss, as shown in Figure 2.

### 3.5. Peri-Implant Soft Tissue

Two implants (2.4%) showed suppuration. Figure 3 displays the descriptive statistics of the bone loss, kertinised mucosa, mBI, and mPI for all the implants. Figure 4 illustrates the distribution of the scores for the probing depth, mBI, and mPI. Accordingly, four implants failed due to an increased probing depth, mBI, and mPI.

### 3.6. Prosthetic Prospective

Among the 48 prostheses, 5 showed prosthetic complications. One was severe, three were moderate, and one was mild. The crown with severe complications was replaced, and therefore was considered as a failure.

Considering the type of prosthesis of the eight failed implants, three were crown abutments, one was bridge abutment, and four were overdenture abutments. All three patients with five implant-supported overdentures had bar attachment. 

### 3.7. Patient Satisfaction

Table 6 shows the median values of the responses to the eleven questions regarding satisfaction. Of these eleven questions, eight showed a median value of five. These median values were tested against the test median value of three using a non-parametric sign and Wilcoxon Signed-rank test. The *p*-values showed that all the responses were statistically higher than the test median value of three. The median value for general satisfaction using a visual analogue scale was 9 out of the maximum value of 10. The mean value for the 11 satisfaction questions was 8.71 ± 1.89. A one-sample *t*-test against the test value of five showed a highly significant difference from this test value (*p* < 0.0001), with the actual value being significantly higher. 

### 3.8. Risk Factors

The medical history of twenty-eight participants (73.6%) showed no medical complications. However, 10 participants (26.3%) had medical diseases, as shown in Table 7. Smoking was declared by five patients (13%), and two failures were associated with smokers. Moreover, six patients (15.7%) had a history of periodontitis and two of them had failed implants. Para-functional habits were recorded for 13 patients (34%), with only 1 related prosthetic failure. 

## 4. Discussion

Few studies have investigated the relationship between surgeon experience and implant survival rates [17]. These authors found that specialist training was not correlated with clinical success; rather, experience with at least 50 implant placements was more indicative of clinical success. Although it may appear logical to correlate experience and failures, many other confounding variables, such as a surgeon’s skills, the case selection, patient compliance, and adapted drilling tools, may also impact the survival rate. The purpose of this outcome audit was to evaluate the influence of the clinicians’ experience on the outcome of dental implants. In addition, it was to identify the associated risk factors that might influence their success and survival.

To increase the degree of clinical validation, a search for evidence of comprehensive criteria for success was performed, but there are not many studies that have looked at different aspects of implant success criteria in one study. Accordingly, the success parameters of this audit were adopted from several studies, as mentioned in the methods. Furthermore, the success rate may seem be low because of applying comprehensive success criteria.

To set the standards for this clinical audit, a systematic review of the survival and success rates of studies with a minimum of 10 years of follow up was considered, with a cumulative mean survival rate of 94.6% and mean success rates from 34.4% to 100% [22]. The 5-year results of Gallucci’s study showed a 95.5% survival rate and 86.7% success rate. Having reflected on these results and the shorter follow up period of our study, we decided on a 95% survival rate and 90% success rate as the predetermined standards for this clinical audit [27]. 

This audit of 84 titanium/Roxoild SLA/SLActive Straumann implants showed an implant survival rate of 100% and implant success rate of 90.5% up to 5 years of follow up. This outcome meets the set standards and even has a survival level well above the predetermined standard. Moreover, these results correspond with previous studies of SLA Straumann implants [5,30,31]. Authors vary in the defining parameters of either the success, survival, or failure of dental implants. Misch et al. [11], in the International Congress of Oral Implantologists, defined the parameters of success, satisfactory survival, compromised survival, and failure differently from Papaspyridakos et al. [26]. Papaspyridakos et al. highlighted the issue that the most of the used parameters in the systematically reviewed literature contain four levels, including implant level, peri-implant soft tissue level, prosthetic level, and patient’s subjective level, as explained in Table 2. Accordingly, increasing this number of parameters may reduce the reported success rates, similar to the findings of Gallucci et al. [27]. They also defined the survival rate of implants and prosthodontics based on the number of failures, by the means of implant removal or prosthesis replacement. Furthermore, Roos et al. [32] highlighted the prognostic criteria of failure when implants were mobile, painful, or infected, or if implants were removed for any reason. The authors also considered implants as surviving if they did not fail. While implants were considered as successful if the marginal bone loss during the first year was <1 mm, there was no more than 0.2 mm resorption annually, no peri-implant radiolucency, and a probing depth of <6 mm on any side (mesial, buccal, distal, or lingual/palatal). All the successful implants were clinically functioning.

Relating clinical findings to the failure rate, two implants presented with acute peri-implant infection, suppuration, and progressive bone loss. One of these implants supported an overdenture prosthesis. This patient lost his teeth due to periodontitis, and he was smoker. In the other implant case with pus discharge, the patient had poor oral hygiene around their implants; it appeared that the patient did not know how to maintain good oral hygiene around the implants. 

The patient satisfaction questions and VAS confirmed that all the patients had an acceptable to high level of satisfaction. This result matched with that of previous studies [33,34]. Nevertheless, three participants were unsatisfied based on the VAS. One of them was unsatisfied because of a fracture of the restoration. The second patient’s expectation was greater than the treatment he received. Functional and aesthetic complications caused the third patient’s dissatisfaction. However, the patients’ answers to the questionnaire questions revealed that the success rate in terms of satisfaction was 100%. Moreover, all the patients showed a high satisfaction rate regarding function and aesthetics.

Anecdotally, during the patients’ interviews for this study at the follow-up appointment, and considering the results of the questionnaire, the majority of patients showed some illiteracy in terms of implant oral hygiene. The oral hygiene instructions in the treatment protocol of the MClinDent programme are usually given verbally. However, the patients’ replies indicated that other methods should be used to convey these instructions. Therefore, the development of brochures to educate patients about the proper care of their implants is strongly recommended [35]. More effort should be applied to ensure that patients fulfil their role in implant maintenance.

The results of this project confirmed the influence of several longevity factors from the literature. It was proven that implant failure is increased in patients who have a history of periodontitis [36,37]. Three out of the eight failed implants in this audit were placed in patients with a history of periodontiits. The patients’ responses for visiting a hygienist or general dental practitioner for maintenance revealed that 50% of the patients with a history of periodontiits never or rarely visited them. Although implant success is reduced with a history of periodontiits, implant therapy in these patients is not contraindicated when there is sufficient infection control and a maintenance programme is provided [38,39]. There is an immediate need to establish an effective method (e.g., written and verbal) for conveying the message of home care. Finally, establishing a recall system might help to conduct a prospective study and more detailed analyses of numerous interesting questions. Thus, another study involving 24 trainees who collectively placed 130 implants demonstrated that the lack of experience of the surgeons did not result in a higher implant failure rate with these implants [40]. It is possible that the presence of an experienced supervisor who assisted with implant planning could explain this favourable outcome; however, this would rely on the assumption that implant failures are the result of inadequate planning, rather than surgical incompetence alone.

The survival rate in this task (100%) matched the dispute that the survival rates for dental implants inserted by trainee surgeons are equivalent to the survival rates stated in the literature. The high success rate of the implants in this project may have been related to the short period of the follow up, carful case selection, and meticulous supervision of the postgraduates within the programme. Moreover, further investigation of the failed implants in this project showed that those cases were not the first cases placed by students. The high success rate of the implants in this audit is not unexpected, since one of the supervisors (consultants/specialist in implantology) usually works as the first assistant.

This clinical audit has several limitations. The first is the low participation rate of patients receiving implant therapy: only 53.5% of the eligible participants were reached and agreed to participate. In addition, human error might have occurred in this investigation. For example, the maximum convexity of the prostheses or an over-hanging crown may have prevented the plastic probe from reaching the base of the sulcus. Knowing that the junctional epithelium around implants is weak could lead the examiner to apply less force during probing. Thus, using a digital probe is recommended. 

## 5. Conclusions

This five-year audit of 84 Straumann implants with an SLA/SLActive surface in thirty-eight partially and fully edentulous patients revealed high survival and success rates (100% and 90.5%) at a mean follow-up time of 26 months. Within the limitation of this clinical audit, regarding implant failure, there were no identifiable contributing factors that were specific to the students’ inexperience. It can be concluded that the implant practice among this trainee programme is satisfactory. A history of periodontitis and lack of patient compliance with supportive periodontal therapy in some cases have been shown to be risk factors associated with increased implant failure, mainly peri-implantitis. 

## Figures and Tables

**Figure 1 healthcare-11-02201-f001:**
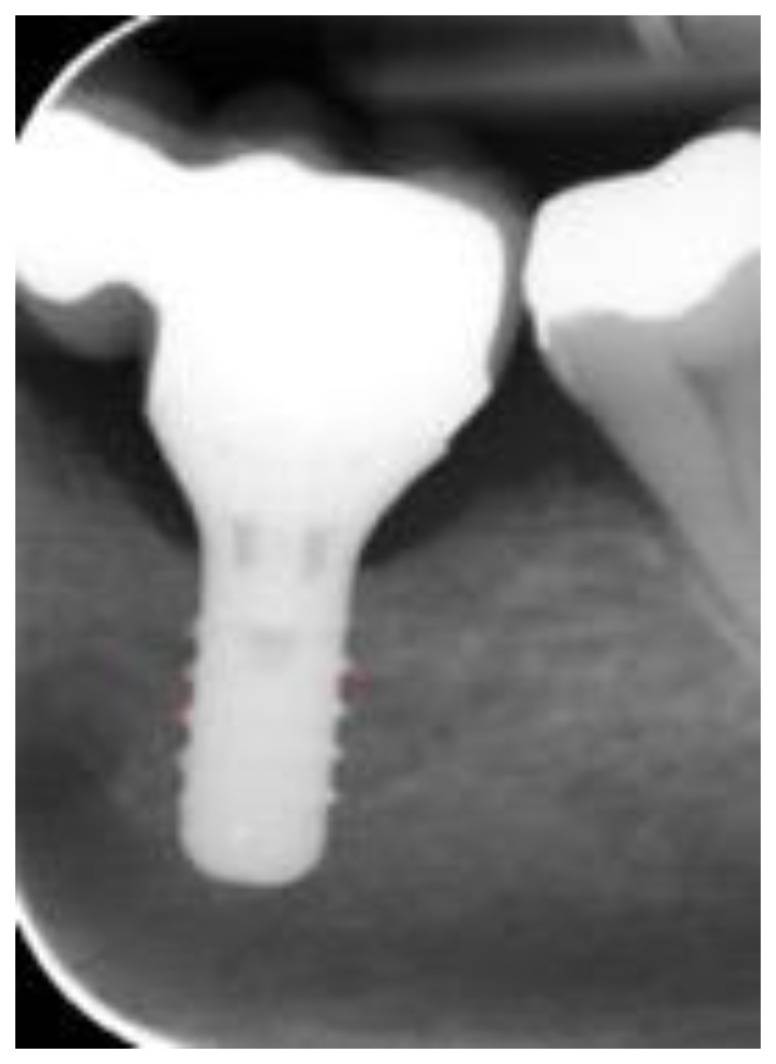
Clear presentation of implant threads.

**Figure 2 healthcare-11-02201-f002:**
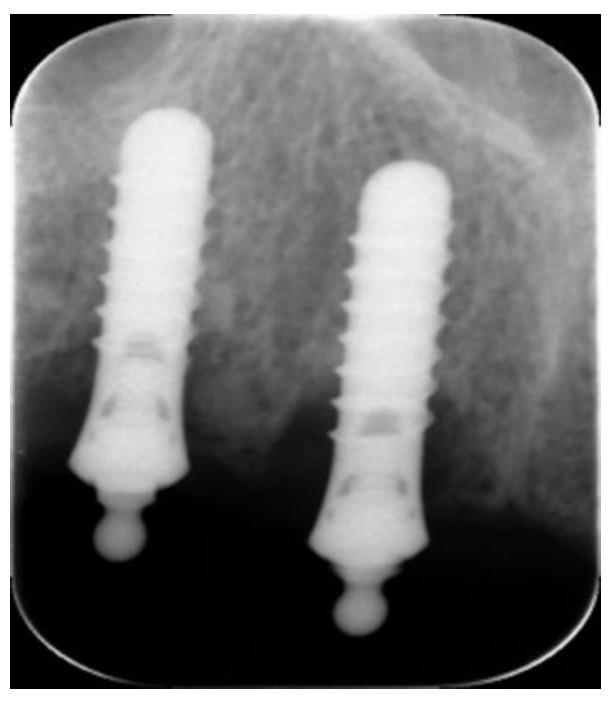
Bone loss on implants retained overdenture.

**Figure 3 healthcare-11-02201-f003:**
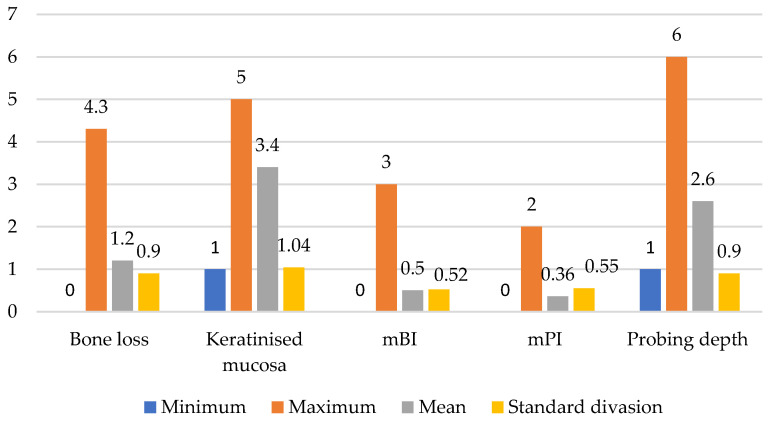
Descriptive statistics of bone loss, keratinised mucosa, mBI, mPI, and probing depth.

**Figure 4 healthcare-11-02201-f004:**
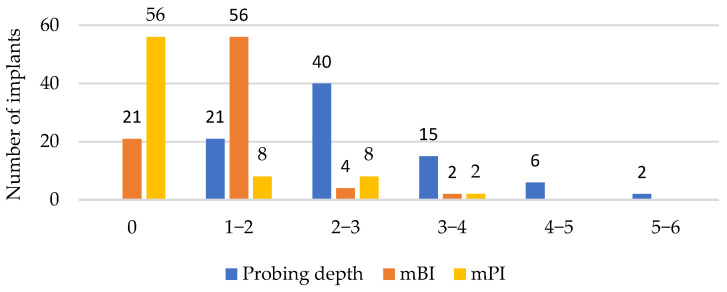
Distribution of scores of probing depth, mBI, and mPI.

**Table 1 healthcare-11-02201-t001:** Questionnaire of satisfaction.

Q1	“My prostheses functions very well, and I can chew on it very well”
Q2	“I feel more secure biting on my implant”
Q3	“To speak, I can use my prostheses very well”
Q4	“I am pleased with the aesthetic results”
Q5	“I can clean my implants very well”
Q6	“My perception of taste has been improved”
Q7	“I need more time to clean my teeth/implant”
Q8	“I got exactly what I expected”
Q9	“I would like this treatment again, if needed”
Q10	“I would recommend this treatment to a friend or relative, if indicated”
Q11	“I often visit my dentist/hygienist for check-up of my implants”

**Table 2 healthcare-11-02201-t002:** Success criteria based on Papaspyridakos et al. [26].

Implant level	Pain Bone loss at first year less 1.5 mm Annual bone loss less 0.2 mm thereafter Mobility Peri-implant radiolucency Infection
Peri-implant soft tissue	Probing depth more than 3 mm Suppuration Swelling (abscess) Bleeding Plaque Index Width of keratinized mucosa more than 1.5 mm Recession
Prosthetic level	Minor complications (chairside approach)—mild Still functioning but need lab repair—moderate Catastrophic failures—severe
Patient satisfaction	Discomfort/paraesthesia Satisfaction with appearance Ability to chew Ability to taste General satisfaction

**Table 3 healthcare-11-02201-t003:** Distribution of implants by the sites.

Site	Number	Percentage
Maxillary anterior	40	47.6
Maxillary posterior	18	21.4
Mandibular anterior	2	2.4
Mandibular posterior	24	28.6
Total	84	100.0

**Table 4 healthcare-11-02201-t004:** Size of the implants.

Size	Number	Percentage
Ø 3.3 NC	8	9.4
Ø 4.1 RC	21	25
Ø 3.3 RN	3	3.6
Ø 4.1 RN	30	35.7
Ø 4.8 WN	22	26.3
Total	84	100

**Table 5 healthcare-11-02201-t005:** Distribution of prostheses.

Prostheses	Number	Percentage
Crowns	44	71
Bridges	13	21
Overdentures	5	8
Total	48	100

**Table 6 healthcare-11-02201-t006:** Questions and statistical tests of the responses using the mid value.

Questions	Median	Sign Test		Wilcoxon Signed-Rank Test	
		Z Value	*p*-Value	Z Value	*p*-Value
My prostheses functions very well, and I can chew on it very well	5	5.167	<0.0001	5.177	<0.0001
I feel more secure biting on my implant	5	5.353	<0.0001	5.325	<0.0001
To speak, I can use my prostheses very well	5	5.676	<0.0001	5.581	<0.0001
I am pleased with the aesthetic results	5	4.167	<0.0001	4.518	<0.0001
I can clean my implants very well	5	5.222	<0.0001	4.869	<0.0001
My perception of taste has been improved	4	4.503	<0.0001	4.503	<0.0001
I need more time to clean my teeth/implant	4	3.298	<0.0001	3.298	<0.0001
I got exactly what I expected	5	3.833	<0.0001	4.291	<0.0001
I would like this treatment again, if needed	5	5.127	<0.0001	4.853	<0.0001
I would recommend this treatment to a friend or relative, if indicated	5	5.029	<0.0001	4.786	<0.0001
I often visit my dentist/hygienist for check-up of my implants	4.5	4.619	<0.0001	4.343	<0.0001
Visual Analogue Scale (VAS)	9	5.029	<0.0001	5.275	<0.0001
Mean	4.24 ± 0.64		t-value = 11.935; *p*-value < 0.0001		

**Table 7 healthcare-11-02201-t007:** Distribution of medical diseases.

Disease	Number of Patients	Percentage
Diabetes Mellitus	2	2.4
Depression and stress	2	2.4
Arthritis	2	2.4
Hypertension	2	2.4
Epilepsy	2	2.4
Smoking	5	13
History of periodontitis	6	15.7
Para-functional habits	13	34

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
