# Peer review of "The Influence of the Clinicians’ Experience on the Outcome of Dental Implants: A Clinical Audit"

_healthcare, 2023, doi:10.3390/healthcare11152201_

Round 1
Reviewer 1 Report
Abstract:
What kind of radiograph exam was requested?
Consider including in the methodology the statistical analyzes carried out.
The follow up period should be considered at the conclusion.
Keywords: implant success; implant survival; and clinician experience must be replaced by terms present in the MeSh terms keywords list.
Materials and Methods:
How many researchers were involved with the clinical analysis of the established parameters? If more than one, was there a calibration between them? Consider the presentation of the kappa coefficient.
Third paragraph: consider structuring in a clear, organized, and easy way for the reader to understand all the methodologies used for each clinical parameter listed.
Table 5 is missing.
3.3 Success was 90.5%, so consider adding the reasons that led to the decrease in this criterion.
Figure 3: Would the vertical scale of the graph be represented by percentage?
The statistical analysis informed was: “Median values were tested against the test median value of 3 using non-parametric sign and Wilcoxon signed-rank test.” However, in figure 3 the mean values are represented specifically, maximum and minimum values, and standard deviation. And the median values?
Information on statistical results is absent, both in the graphical representations and tables, and in the text, which makes scientific discussion unfeasible.
Legends need to be added for better understanding.
Reviewer 2 Report
Comments are in the pdf

Comments are in the pdf
Round 2
Reviewer 1 Report
Dear authors, congratulations for your study. The reviews are adequate, so I'm considering this study for publication.
About the legends: all figures and tables: all figures and tables could be accompanied by caption addressing specific and detailed information.
Author Response
Thanks for your comments. I reviewed the table and figure once again.